# Exploratory Qualitative Study to Investigate Factors Influencing Men’s Utilization of Sexual and Reproductive Health Services in Kwa-Zulu Natal

**DOI:** 10.3390/ijerph21121632

**Published:** 2024-12-08

**Authors:** Mpumelelo Nyalela, Thembelihle Patricia Dlungwane

**Affiliations:** School of Nursing and Public Health, College of Health Sciences, University of KwaZulu-Natal, Durban 4041, South Africa; dlungwane@ukzn.ac.za

**Keywords:** sexual and reproductive health services, men’s barriers and facilitators to reproductive health services, reproductive health service utilization, explore men’s health services, family planning, circumcision

## Abstract

Sexual and reproductive health (SRH) is essential for men’s health, but a large body of research has indicated that the underutilization of most SRH services by men is a persistent issue that needs to be addressed. Men’s reluctance to access sexual and reproductive health services is one of the factors that leads to high morbidity and mortality rates among men, although their diseases may have been prevented or cured. This study aimed to explore factors that influence the decision of men who resided in men’s hostels and who accessed urology clinics in KwaZulu-Natal to seek help for their sexual and reproductive health issues. An exploratory qualitative approach was adopted using focus group discussions. We interviewed seventy-two men of ages above 15 years. The data were analyzed thematically. The Biomedical Research Ethics Committee (BREC) of UKZN granted ethical clearance (BE 347/19). Of the 72 interviewed men, thirty-three men attended urology clinics in the selected hospitals, and thirty-nine men resided in the hostels around Durban in KZN. Seven themes (lack of awareness of SRH services; participants’ reluctance to access SRH services; influence of culture and religion; lack of financial resources; influence of relationship dynamics; perceived low risk of individual sexual behaviors; and healthcare factors that discourage men from accessing SRH services) emerged from the data that were identified as barriers to SRH service utilization by men, whilst three themes (healthcare enabling factors; access to general information on SRH services; and personal motivational factors) emerged as factors that encouraged the participants to access these services. The participants’ reluctance to access SRH services was attributed to the lack of awareness of available SRH services, the influence of culture and religion, lack of financial resources, relationship dynamics, the perceived low risk of sexual behaviors for individuals, and healthcare workers’ negative attitude towards men requiring SRH services. The availability of healthcare resources, the appointment of more male healthcare workers, and more positive attitudes among healthcare workers will encourage men to utilize SRH services. The exposure of various barriers to SRH service utilization by this investigation warrants urgent attention from the government to impart knowledge about this phenomenon to men.

## 1. Introduction

Utilizing sexual and reproductive health (SRH) services is fundamental for the well-being of men and averts high mortality and morbidity rates among them at a premature age [1,2,3]. Men require SRH services to inform them of and support them with issues such as family planning, vasectomy, sexually transmitted infections (STIs), Human Immunodeficiency Virus/Acquired Immunodeficiency Diseases (HIV/AIDS), infertility, male cancers, and treatment of sexual anomalies. However, the underutilization of most SRH services remains prevalent and is well documented [4,5,6,7,8,9]. Previous research has indicated that most men underutilize SRH services because they regard reproductive health services as a space reserved for women [10].

According to the United Nations Population Fund (UNFPA), SRH is essential for well-being, encompassing safe and satisfying sexual experiences, reproductive capabilities, and informed decision-making [11]. However, men’s access to SRH services is hindered by masculine norms, cultural beliefs, and barriers such as long wait times, negative staff attitudes, lack of men-friendly services, and concerns over privacy [4,5,8,12,13,14,15,16,17]. Many men lack information about SRH services, and procedures like male sterilization are rarely discussed [18]. Additionally, poor infrastructure, unclear policies, and healthcare workers’ limited capacity contribute to men feeling unwelcome in SRH services [19].

In most African countries, male participation in sexual and reproductive health support services has proved to be challenging, especially where there are culturally defined gender roles and manifestations of masculinity [1,9,12,13,16,19]. Furthermore, the lack of coordinated and comprehensive guidelines and policies to improve SRH services for men and boys is still sorely felt in many developing world contexts [4]. The literature argues that men seldom access sexual and reproductive health services and that this abstinence may possibly lead to high morbidity and mortality rates regardless of the fact that most of these illnesses are curable [1,2,5,16].

The current study was conducted among men recruited at urology clinics and men-based hostels near Durban in KwaZulu-Natal (KZN). We aimed to explore factors that influence the decision of men who reside in hostels and who access urology clinics in KwaZulu-Natal to seek help for their sexual and reproductive health issues. This study was undertaken to further unpack the findings of an earlier quantitative phase of the larger study, during which it was found that age, level of education, and employment status were significantly associated with SRH service utilization, or the lack thereof [20].

## 2. Material and Methods

### 2.1. Study Design

This study employed an exploratory qualitative approach. A qualitative design was chosen in order to gain further insights and elicit detailed descriptions of the phenomenon under study.

### 2.2. Study Setting

A study setting is where this study is conducted, and information is gathered. The data were collected by means of FGDs at four urology clinics and four men’s hostels in the Durban area in KZN. Geographically, KZN is situated along the east coast of South Africa. KZN has the second largest population, with an estimated 12.34 million people (19.6%) living in this province [21].

Urology clinics exclusively treat men and women with sexual and reproductive health problems. They are located on hospital premises and use rooms that are separate from other clinics so that only patients with SRH problems are consulted at these facilities. At the clinics, quiet consultation rooms in the vicinity of the main waiting areas were used for the FGDs, while the halls in the men’s hostels were used for this purpose. These halls are commonly used for community meetings but were made available for the FGDs during this study.

### 2.3. Study Participants

Study participants included men who were already utilizing SRH services and those who had the potential to utilize SRH services and resided in hostels in large numbers. All men who were 15 years and older and utilized or could potentially utilize available SRH services were recruited. Males younger than 15 years were not recruited. Moreover, men whose conditions would make it impossible to participate were not recruited. For instance, men who were seriously ill and in pain and men in hospital wheelchairs and stretchers were not recruited. Among the recruited participants, in both hospital and hostel settings, some of the participants were unable to partake in the interviews for various reasons. For instance, some participants who were at the front of the queue were not willing to leave their posts and join discussions. In hostels, some men could not be part of FGDs and alluded to urgent matters they had to attend. We interviewed seventy-two men, which included thirty-three men who attended urology clinics in the selected hospitals and thirty-nine men who resided in the hostels around Durban in KZN.

### 2.4. Sampling Strategy

In both settings, men were conveniently selected. In the hospitals, the Operational Managers (OMs) approached potential participants while they were waiting in queues to be seen by doctors. These individuals were informed of the purpose of this study and were requested to volunteer to join the group. In the hostels, the various managers (Izindunas) recruited a group of men based on their availability, and they were also informed of the purpose of this study. FDGs of 8 to 12 participants were formed in each of the targeted settings. The size of the groups varied slightly, and each discussion was conducted in a quiet venue where no interruptions were experienced.

### 2.5. Data Collection

Focus group discussions (FGDs) were used as the data collection method, and a semi-structured interview guide was used to guide the face-to-face interviews until data saturation had been achieved. The data were collected between May and June 2022.

We introduced ourselves and thanked the interviewees for their time and willingness to share their views. We briefly went over the purpose of this study and the scope of the interview prior to commencing the interviews. The interview questions and the consent process were explained to the participants, and they were asked to raise any concerns or questions they might have.

Men who agreed to participate were given consent to sign. In the hospitals, Operational Managers (OMs) arranged private and quiet rooms to conduct focus group discussions. In men’s hostels, managers (Izindunas) organized men and gathered in hostel halls. In South Africa, “hostel” refers to a single-sex barrack, dormitory, or housing compound for the accommodation developed and designed for black migrant workers in urban areas in the early days of South Africa’s history.

All FGDs were conducted by the principal investigator (PI) and a trained research assistant (RA). Interviews were conducted to gain deeper insights into the responses emanating from questionnaires. The authors developed the interview guides to meet the aim of this study from previously published studies investigating factors that influenced the decision of men to seek help for their sexual and reproductive health issues. When developing our semi-structured questionnaires, we considered the goals and research focus. We then developed a list of general questions that we wanted to ask during the interviews. The questions that did not relate to a research goal were deleted to ensure that each question was relevant to the goals of the research project. We avoided using long or complex questions. Instead, probes were used to obtain more details and clarify responses. We limited the guide (Table 1) to one page so that it was easy to refer to and to make sure that we did not achieve a response level that was too low. Creating such a guide can help to focus and organize the line of thinking and, therefore, questioning.

The participants were asked open-ended questions to allow them to respond frankly about their views on men’s willingness or reluctance to utilize SRH services. The interviews took 45–60 min each and were tape-recorded and transcribed verbatim. Field notes were taken during each interview as this provided key points for further probing. During the data collection, participants’ anonymity was maintained by coding data with letters from A to L, their ages in brackets, and the first letter in the name of the study setting, for example, A(45)–G.

To solicit rich data, we used an informal conversational strategy. We used the interview guide throughout each interview to ensure that the dialogue remained focused on sexual and reproductive health service utilization. If a question drew a blank stare, we would reframe the question to make it clearer and tie it to the participants’ earlier comments. At the end of each interview, we would summarize key ideas and themes back to the participants to ensure we had a proper understanding of their meaning. At that point, participants were able to provide clarification of prior statements and additional information about their experiences. The PI administered the questionnaires in all the interviews to reduce interviewer bias and to limit variation in interview technique. However, the RA intermittently posed follow-up probes to further explore points as they arose during the interview.

The researchers ensured that questions were asked in as neutral a form as possible and that interviews took place in settings comfortable for the researcher and the respondent to allow the researchers to be sensitive to respondents. The interviews were conducted in an emphatic and conversational style. However, the researchers tried to ‘bracket’ or set aside their assumptions and theories and instead focused on the research participants’ points of view and their unique lived experiences. To validate the interview guide, the researchers conducted a pilot study to test the interview questions, as well as the interviewing style and approach. There were no changes to the data-collection instrument because of the pilot study.

### 2.6. Data Management

The digital voice recorder containing the data was stored under lock and key. Audio files were kept safely on a password-protected computer and transcribed verbatim within 24 h after each interview. The data were stored in a secure place accessible only to the PI and the supervisor. PI is accountable for the proper maintenance and availability of the primary research collected for this study, which will be maintained for five years, after which it will be destroyed in line with the University’s research data management policy. The report of the findings will be submitted to the School of Nursing and Public Health and Provincial Department of Health.

### 2.7. Data Analysis

Researchers used a tape recorder and then transcribed the tapes into written textual data. Any interruptions to the interview (INTERRUPTED), silences (SILENCE), or a pause in the interview (….) were indicated in the text. The RA transcribed the data, and the PI and supervisor double-checked the transcripts for completeness, accuracy, and clarity. The translated data were cross-checked with the audio file to ensure proper transcription and translation. The PI and the supervisor read the translated data repeatedly to understand the concept and related meanings of the data. The manual approach to organizing data was utilized. The data were imported into a spreadsheet for further analysis.

The data were analyzed as follows: (1) listening to the audio tapes before transcription; (2) transcribing the audio tapes; (3) inductively assigning codes to text segments; and (4) categorizing the data by coding and identifying themes. The PI carried out coding, categorizing, and theming. During this stage of the analysis, PI began by carefully reading the transcripts. Each transcript was read multiple times, actively observing meanings and patterns that appeared across the data set. Once familiar with the data, we began assigning codes to various excerpts arising out of the transcripts as we read them. We created a table in the Excel spreadsheet to organize coded data. As we read through the data again, interesting excerpts were identified and applied to the appropriate codes. Excerpts that represented the same meaning had the same code applied. New codes and excerpts were also identified as we read along. The excerpts associated with a particular code were grouped together. Codes with similar meanings were then merged, and excerpts were grouped together and organized under potential themes.

The initial set of themes was reviewed and revised to ensure that each theme had enough data to support them and was distinct. Similar themes were merged and formulated in such a way that they could come together into a narrative. The supervisor checked and verified key themes, and common categories were agreed upon. The themes overlapped and emerged either as barriers or facilitators of SRH service utilization.

### 2.8. Trustworthiness and Rigor

The trustworthiness of this study was established by ensuring that it encompassed the four characteristics of trustworthiness and rigor.

Credibility alludes to confidence in the truth of the data and the interpretation thereof [22]. The data were recorded, and notes were taken simultaneously to ensure proper capturing of data. Researchers performed an extensive literature review and examined the previous research findings during all phases of this research. The researchers had prolonged engagement with the participants in all the research sites during the research period. Prior to data collection, researchers visited research settings for gatekeeping permission and, most importantly, to build rapport with the potential participants. On the days of data collection, researchers would normally arrive early and chat with participants to make them feel comfortable disclosing information. Researchers were mindful of their preconceptions to avoid influencing data collection. The researcher reflected on the impact of their role, personal background, and culture before, during, and after this study to prevent and avoid the occurrence of any possible biases. The PI debriefed the RA about the data collection and data transcription to help uncover taken-for-granted biases, perspectives, and assumptions. During data collection, member checking was ensured by summarizing key ideas and themes back to the participants to ensure that we had a proper understanding of their meaning.

Transferability refers to the ability to apply the findings in other contexts or to other participants [22]. The researchers employed purposive sampling in selecting participants whom we knew would provide rich and relevant information pertaining to this study. The researcher described the context/setting, the research process, and the findings in detail to enable readers to decide on the applicability of the current study to their context.

Dependability refers to the provision of evidence such that if it were to be repeated with the same or similar participants in the same or similar context, its findings would be similar [22]. Researchers documented real-life experiences so that this study’s findings were based on the experiences and ideas of the participants rather than on the researcher’s beliefs, preferences, and assumptions. Personal involvement at the site with the participants through participant observation provided an opportunity to experience the insight necessary for detailed descriptions, which are imperative in qualitative data. The research method is fully described. A tape recorder was used to increase reliability during all interviews. Dependability was ensured by requesting an external audit from the qualitative researcher who was not involved in the research process to examine both the process and product of this research study. The purpose was to evaluate whether or not the findings, interpretations, and conclusions were supported by the data.

Confirmability refers to the potential for congruency of data in terms of accuracy, relevancy, or meaning [22]. Researchers ensured this by substantiating the report of the interviews and reviewing similar studies that had previously been conducted to determine if the participants’ responses matched the views expressed in the literature. The researcher ensured accuracy through field notes and tape recording. There was a consensus of agreement on the findings based on the data analysis between the PI and the supervisor. Furthermore, researchers requested the qualitative expert to challenge the process and findings of a research study by looking at findings to assess the adequacy of data and preliminary results. This important feedback led to additional data gathering and the development of stronger and better-articulated findings. The researcher clearly described each stage of the research process, exploring and justifying what was achieved and presenting reasons for the decision taken. The researchers ensured the safekeeping of the recorded tape cassettes, written documents, and notes from the interviews.

### 2.9. Ethical Considerations

Ethical clearance (BE 347/19) was granted by the Biomedical Research Ethics Committee (BREC) of UKZN on 6 October 2020. The KZN Provincial Health Department approved the data collection sites and allowed the research team access to patients in the targeted healthcare facilities, while gatekeepers’ letters were obtained from the hostel managers. All the participants signed the consent form before data collection to indicate their willingness and voluntary participation in this study.

## 3. Results

Table 2 presents the socio-demographic characteristics of the respondents. The 72 respondents comprised 39 (54.2%) males who resided in the hostels and 33 (45.8%) males who attended urology clinics. The respondents were predominantly IsiZulu speaking (65.3%), as most people living in KwaZulu-Natal were members of the Zulu ethnic group. Most (68.1%) of the respondents were above the age of 44, which was indicative of the demand by a more mature group for the use of SRH services. Most of the respondents were single (39 or 54.2%), had a pre-secondary school education (46 or 63.9%), and were unemployed (38 or 52.8%).

Table 3 presents themes emanating from the participants. Emerged themes, as discussed below, were identified as either barriers or facilitators to SRH service utilization by men.

### 3.1. Barriers to the Utilization of SRH Services

Seven themes emerged from the data that were identified as barriers to SRH service utilization by men. These themes are the following: (1) lack of awareness of SRH services; (2) participants’ reluctance to access SRH services; (3) influence of culture and religion; (4) lack of financial resources; (5) influence of relationship dynamics; (6) perceived low risk of individual sexual behaviors; and (7) healthcare factors that discourage men from accessing SRH services (Table 3).

#### 3.1.1. Lack of Awareness of SRH Services

The participants were unaware of SRH services for men that were available in health establishments other than condom use, medical male circumcision (MMC), and treatment for STIs and HIV. They attributed their underutilization of a circumcision service to the lack of awareness about such a procedure. Moreover, most participants did not know about modern SRH services, such as infertility support.


*Mr. C(44)-J: “The only services I know that are available in the hospital are things like condoms, circumcision, HIV, and STIs.”*



*Mr. D(45)-N: “Sometimes you want fertility tablets as you are sterile. When you need help, you don’t know where to go.”*


Some participants saw no reason for undergoing circumcision, claiming that they cleaned themselves properly. They, therefore, demonstrated a lack of awareness of health hazards associated with non-circumcision, such as contracting HIV and other STIs.


*Mr. B(61)-Gr: “I don’t see any reason I should consider circumcision because I can retract my foreskin and clean my penis properly.”*


#### 3.1.2. Participants’ Reluctance to Access SRH Services

Most participants felt that it was not important to undergo health screening when they were not feeling sick, and all believed that, as men, they had to be strong and show endurance. Furthermore, they revered their masculinity and argued that health screening was ‘women’s business’ and unmanly. The participants’ hesitancy to access SRH services was illustrated when they stated that they waited until their sicknesses worsened before seeking help for any SRH problems, hoping that the symptoms would disappear.


*Mr. F(24)-S: “As men, we always believe that illness will go away. We wait until the situation has worsened because we want to seem strong.”*



*Mr. B(55)-S: “As men, we always think that screening and check-ups are for women as they are the ones giving birth.”*


The participants’ reluctance to utilize SRH services was also demonstrated when they reported non-condom usage, claiming diminished sexual pleasure and regular loss of erection while wearing a condom. One participant equated wearing a condom to eating a wrapped sweet. The participants also demonstrated little interest in using condoms for family planning purposes, yet they risked contracting infections and claimed they had ways of avoiding impregnating women.


*Mr. H(82)-G: “I don’t use a condom just like I don’t eat sweets that are still wrapped in paper.”*



*Mr. B(52)-K: “Eiy! I tried to use a condom, but I lost erection.”*



*Mr. F(42)-W: “I don’t use condoms but practise pre-ejaculation withdrawal to prevent impregnating a woman.”*


Most participants felt that utilizing services such as vasectomy was inappropriate as they worried about their consequent inability to produce more children in the event of losing a member or more members of the family.


*Mr. E(58)-K: “If I undergo vasectomy, it means I am castrated.”*



*Mr. H(82)-G: “If my children die at the same time, how will I start a new family if I am castrated?”*


The participants asserted that, due to fear of adverse effects such as delayed wound healing, infection, and possible penile amputation, they were reluctant to undergo circumcision.


*Mr. F(24)-S: “I would like to undergo circumcision, but I worry about the complications, such as delayed wound healing that can result in my penis being cut off.”*


#### 3.1.3. Influence of Culture and Religion

Another factor that contributed to the unwillingness of the participants to access SRH services was the influence of their culture on their beliefs. Most participants preferred to consult traditional healers instead of going to a clinic or hospital for their sexual health problems. Although some participants were knowledgeable about the importance and benefits of circumcision, they were reluctant to undergo this procedure, asserting that it was not part of their culture.


*Mr. B(41)-G: “It is our culture to treat SRH issues with traditional medicines.”*



*Mr. E(59)-Gr: “I don’t believe in circumcision. I heard it is more like the culture of certain nations, such as Xhosas.”*


#### 3.1.4. Lack of Financial Resources

The participants argued that a lack of financial resources was a deterrent to SRH service utilization, particularly as most missed their appointments due to a lack of money for public transport or taxis.


*Mr. I(29)-S: “I am from Mtuba [Northen KZN] and I am using money for trips to travel. I am struggling and don’t have enough money, so I miss some of my doctor’s appointments.”*


Some participants opt to use private health services because public hospitals only dispense medication for a short period, and it often runs out before the next appointment.


*Mr. E(52)-S: “Tablets are sometimes unavailable in the hospital and tablets are expensive when we go to a private pharmacy.”*


However, they affirmed that private doctors cost a lot of money.


*Mr. A(45)-J: “The reason why we are not going to the doctor is that we have to pay for the consultation and the tablets, but we don’t have enough money.”*


#### 3.1.5. Influence of Relationship Dynamics

Some participants reported that their female partners regularly refused to use condoms, claiming that they caused side effects such as a rash and pain that resulted in sexual frustration.


*Mr. G(78)-N: “Sometimes, we are forced not to use condoms because our ladies complain about sores after having sex using a condom.”*


The participants also argued that they could not be tested for HIV because they feared losing their partners should they learn that they had contracted the virus due to unfaithfulness.


*Mr. F(37)-G: “As men, it’s not easy to check for HIV because we know where we have been most of the time.”*


#### 3.1.6. Perceived Low Risk of Individual Sexual Behaviors

Some participants did not utilize certain SRH services as they argued that they were not at risk of contracting infections due to their sound sexual behaviors. Age emerged as a key factor that determined whether or not men would access SRH services such as circumcision. The older participants did not see the need for circumcision, stating that they were too old and no longer sexually active.


*Mr. C(83)-L: “I have not had circumcision because I see no reason as I am old and not sleeping around.”*


The low-risk mindset was also associated with various beliefs such as trust in partners; self-trust due to good health, a good immune system, and prevention skills; traditional medicines; the support of the ancestors; and faith in God. Low condom usage was, thus, prevalent among the participants, especially those who were in long-term relationships.


*Mr. D(79)-K: “No, I don’t use a condom because I trust my wife.”*



*Mr. H(57)-K: “I don’t believe in condom use because I use my traditional medicines.”*



*Mr. E(58)-K: “I don’t need to use a condom because I trust God will protect me.”*


Being too trusting resulted in some participants using a condom only during the first or second sexual encounter (either the first round of sex or on the first day of a relationship), and then most stopped using them on subsequent occasions.


*Mr. H(32)-S: “We men are very quick to give our trust. Some of us only use a condom in round one; in the second round, we don’t use it.”*


#### 3.1.7. Healthcare Factors That Discourage Men from Accessing SRH Services

Major barriers to men’s willingness to access SRH services that were mentioned were that the staff might share their confidential information with others and the fact that some staff members would shout at them in public and ridicule them for being diagnosed with an STI. Healthcare workers’ negative and judgmental attitudes, as well as a lack of privacy in clinics, caused most men to avoid visiting these facilities even if it seemed necessary to be tested for HIV.


*Mr. G(78)-N: “Another cause for the lack of HIV testing is attributable to the nursing staff. They make it more difficult to go for an HIV test as they will stand and shout in the corridors, saying that if you want to have an HIV test, you need to come to this side. They call your name out loudly in public so even if someone had the intention to be tested, they withdraw because of the shouting and exposure.”*


Most participants attributed the underutilization of SRH services to the lack of confidentiality, privacy, respect, and the rudeness of young healthcare workers. The participants felt that young healthcare workers ignored them and walked up and down the passage while they had to sit in long queues without being helped.


*Mr. F(72)-K: “A challenge is the lack of respect by young female nursing staff. You share your problem with one of them, and in no time, they ridicule you by discussing you with other nursing staff. You also see them moving up and down not knowing what they are doing, while we are waiting for their assistance.”*


When asked how culture influenced their decision to utilize SRH services, most participants indicated that culture did not prevent them from using SRH services. Instead, they attributed their reluctance to attend clinics to the judgmental attitude of nurses. The participants also pointed out that when they grew up, they often went to clinics, but now that they were adults, healthcare workers ill-treated them.


*Mr. A(71)-G: “Culture does not influence whether we utilize SRH services or not. We grew up going to the clinic to get immunizations and did not know anything about traditional medicine. The problem starts when you are an adult! The nurses insult us and that’s why we go to traditional healers.”*


Apart from resenting nurses’ negative attitudes, the participants also detested the long queues and hours they had to wait before being attended to. They also disliked the long distances they had to travel to health facilities and cited this as a major barrier to their willingness to utilize SRH services. They admitted that they would either stay at home or consult traditional healers, as they argued that no humiliation was experienced when consulting such healers. They also stated that traditional services were quick and that their privacy and confidentiality were never compromised.


*Mr. A(71)-G: “Before we go to the clinic or hospital, we think about being insulted by nurses and we end up not going.”*



*Mr. B(45)-W: “It is expensive to consult traditional healers when you are not working; however, we still consult them or stay at home rather than go to the hospital for our SRH problems as the nurses are very rude.”*



*Mr. C(39)-W: “Initially, we consult traditional healers because of nurses’ bad attitude in the hospital. If the traditional healer can’t solve my problem and the situation worsens, then I go to the hospital.”*


Most participants felt embarrassed when they were treated by female healthcare workers and found it difficult to open up about their conditions, such as sexually transmitted infections and sexual dysfunction.


*Mr. C(44)-J: “The challenge we have as men is that there are many female nurses when we get to the clinic. Sometimes it’s difficult to share your SRH problems, such as gonorrhea, with the female nursing staff.”*


The participants were also frustrated and disappointed by the health system’s lack of SRH services. Furthermore, those residing in hostels complained about the unavailability of health establishments near their places of stay, while such facilities were easily accessible to those living in urban areas. They further reported that they had to travel long distances to access health services in urban areas and that this was expensive.


*Mr. B(75)-N: “The nurses told me that cancer screening results are usually not returned. Nonetheless, they advised me to be persistent and regularly check in. Although I did so, I have not received my results till today.”*



*Mr. A(71)-G: “We do not have a clinic in our area. We use taxis and buses to reach the clinic and transport is very expensive.”*


### 3.2. Factors That Facilitate SRH Service Utilization

Despite the factors that emerged as barriers to SRH service utilization, this study also revealed several factors that encouraged the participants to access these services. During the FGDs, the participants admitted that they were reluctant to go to a clinic or hospital at the onset of their condition. The men who were interviewed at the hospitals also indicated that men tended to wait until their problems became severe. However, the factors that encouraged healthcare visitation were as follows: (1) healthcare enabling factors; (2) access to general information on SRH services; and (3) personal motivational factors (Table 3). 

#### 3.2.1. Healthcare Enabling Factors

The participants agreed that talking to a male nurse was easier than talking to a female nurse. They perceived most female nurses as rude and argued that they compromised their privacy and confidentiality. The participants further stated that men understood one another and that they were comfortable sharing their sexual health problems with male nurses. The presence of male nurses in health establishments usually motivates them to go to hospitals or clinics when experiencing sexual health issues.


*Mr. F(37)-G: “We want more male nurses in hospitals and clinics. It is better to be treated by other males because we understand each other. When more male nurses are employed in hospitals or clinics, we shall be motivated to visit these health establishments.”*


#### 3.2.2. Access to General Information on SRH Services

Although circumcision was not part of most men’s culture, learning about post-circumcision benefits, such as cleanliness and preventing diseases/STIs, encouraged some participants to consider circumcision. Learning about the benefits of MMC enabled the participants to comprehend the rationale for undergoing this procedure. Some also reported that they used condoms to prevent contracting diseases.


*Mr. H(32)-S: “I have undergone circumcision because it is right. It helps avoid the collection of dirt under the foreskin. Many diseases are caused by the collection of dirt under the foreskin. When you have done circumcision, the penis is always clean.”*



*Mr. B(75)-N: “The condom is very safe, even if you have other STIs like cauliflower, it won’t affect the female partner.”*


#### 3.2.3. Personal Motivational Factors

Participants in the focus group discussions repeatedly mentioned that removing the foreskin enhanced sexual performance and that this motivated them to undergo circumcision.


*Mr. A(58)-N: “I had circumcision because women get more sexual pleasure when we have done circumcision.”*


Participants who were careful about their lifestyle and in stable and mutually faithful relationships reported being regularly tested for HIV.


*Mr. D(45)-N: “I tested for HIV because I am not a promiscuous man.”*


## 4. Discussion

The aim of this study was to explore factors that influence the decision of men who reside in hostels and who access urology clinics in KwaZulu-Natal to seek help for their sexual and reproductive health issues. The findings suggest that multifaceted factors influence men’s decisions to utilize SRH services. A lack of awareness of the significance of SRH services was revealed when most participants admitted that they were not eager to use family planning methods such as vasectomy and condoms despite their high efficacy and low-risk benefits. Similar to a study conducted in the Philippines, most participants preferred conservative methods such as withdrawal/coitus interruptus and calendar guidance to avoid impregnating women [23]. However, the literature states that withdrawal (or coitus interruptus) is not 100% effective as it has an approximate 4% failure rate [24].

The men perceived family planning as the responsibility of women as it is they who fall pregnant. Participants attributed their lack of awareness of SRH issues to the deficiency of health education of men who visit various health establishments. This suggests that men must be motivated to utilize SRH services by involving them in educational health talks, as the limited information they receive may contribute to their inability to make informed decisions about accessing SRH services.

Most participants were reluctant to be regularly screened for diseases as they associated such services with women’s health. They also felt that it was unnecessary to be exposed to health screening when they were not sick. This finding resonates with a study that was conducted in Mexico that also revealed that men were reluctant to use certain SRH services unless they experienced serious health problems. It, therefore, seems that most men tend to wait until their conditions are severe before accessing healthcare establishments [12,25,26].

Moreover, some participants chose not to undergo circumcision as they felt that it was not part of their culture. This phenomenon is not unique to this study, as other studies have also suggested that men tended to be reluctant to undergo circumcision as it was not part of their culture [27,28,29]. In one study, the participants termed it “an alien culture” [30]. The influence of culture was also evident when the men admitted that they preferred it when traditional healers dealt with their sexual health problems. This finding is corroborated by other South African studies that also state that traditional medicine is preferred over formal healthcare treatments [12,31,32,33]. Some studies have cited social and cultural norms and stereotypical beliefs that men did not need health services as obstacles that might hinder men from accessing SRH services [34,35,36]. Apart from cultural beliefs, limited financial resources also indirectly hinder access to SRH services. Most men residing in remote and rural areas do not have the funds to pay for their transport and struggle to meet their appointment dates consistently. Various scholars have affirmed that cost implications and distance to health facilities restrict access to SRH services [17,28,31,32,36]. The participants also highlighted their inability to purchase medicines to supplement those that ran out before the next appointment.

As affirmed in other studies, a further hindrance was men’s assertion that female partners are to be blamed for men’s aversion to condoms as they complained about post-coitus pain and infections. For instance, in a Laos study, female participants complained about side effects after condom use, such as rash, pain, and diminished sexual pleasure [23,37,38]. Furthermore, participants asserted that they were reluctant to undergo a test for HIV and feared losing their partners due to STIs or an HIV-positive status. As reported in other studies, participants admitted that they were embarrassed and humiliated when they had to be screened for STIs, and they also worried that they would lose their partners or that potential partners would avoid contact with them [32,33,39,40].

It is evident that people’s denial of the fact that they may contract STIs causes them to underutilize SRH services. For example, older men believe they are not at risk of contracting HIV because they are too old and trust their partners [32,33], while younger men believe in their masculinity and invulnerability and claim that they are absolved from contracting diseases as they trust their partners and use orthodox contraceptive methods such as traditional medicines, the support of the ancestors, and God’s mercy. Echoed in other studies, trusting too soon was apparent when some men only used condoms during the first or second sexual encounter and discontinued using them on subsequent occasions [31,41].

As affirmed in other studies, most participants also argued that female healthcare workers’ negative attitudes toward men with SRH issues made it difficult for them to seek help, and, therefore, men tended to seek assistance from traditional healers, particularly if they were not assisted by male healthcare workers [13,25,36,42,43]. Men are comfortable with male healthcare workers when they have to discuss sensitive health matters as they are worried that female healthcare workers might breach their privacy and confidentiality [4,8,19]. In addition, other studies also indicated that waiting times, poor treatment by clinic staff, and being seen by members of the community at a clinic were cited as factors that contributed to men’s decisions not to seek SRH services at public health facilities [13,37,40,41,44]. These findings suggest that strategies to create a male-friendly environment at clinics for men must be implemented to encourage men’s utilization of SHR services.

The findings of the current study also revealed several factors that facilitated SRH utilization. Despite men’s reluctance to utilize SRH services, the presence of male nurses in health establishments may motivate more men to access facilities where it will be possible to speak to and be assisted by male nurses. A study conducted in Cape Town further posited that awareness of the benefits of utilizing SRH services, such as condoms, encouraged some men to use them to prevent contracting diseases [41]. In addition, awareness campaigns that informed communities of the benefits of circumcision and personal motivational factors inspired some men to undergo circumcision even though this procedure was against their cultural beliefs. Despite the paucity of studies indicating the influence of cultural beliefs on the utilization of SRH services such as MMC and vasectomy services in KZN, this divergence of men’s perceptions is echoed in fewer studies. For instance, some Zulu men rejected undergoing MMC, viewing it as a sin, unnatural, unknown, and unnecessary [45]. Conversely, another study found that some men accepted MMC, linking it to the affirmation of male sexuality and masculinity [46]. The divergence of men’s sentiments on the influence of cultural beliefs indicates that such beliefs do not unilaterally influence SRH service utilization.

The influence of cultural and religious beliefs on the utilization of SRH services is not unique to KZN men. Several studies revealed that a threat to men’s masculinity, viewing MMC as an alien culture, part of a foreign religion, or tampering with God’s creation hindered them from undergoing it [23,47,48,49]. Furthermore, the mere discussion of sexual matters is deemed taboo by most cultural and religious practices, hence hindering several men from accessing SRH services [35,43,47,50]. In a study conducted in KZN, the cultural norm that men did not discuss their sexual matters with women was highlighted as a deterrent to accessing SRH services [51]. Moreover, cultural beliefs still highly influence fertility and contraceptive use by men, resulting in low utilization of fertility services [43], especially in communities experiencing poor socio-economic status [34,52]. The influence of cultural beliefs also has an impact on men who delay accessing SRH services, such as cancer treatment, claiming the need to consult their ancestors prior to the procedures [51]. Moreover, some studies revealed that some men rejected vasectomy, claiming the cultural need to produce until they die and that vasectomy was the same as castration [30,31,32,33]. Therefore, ongoing awareness campaigns to inform the public of the benefits of SRH service utilization and the appointment of more male nurses in public health facilities can encourage men to access such services more readily and regularly.

Our findings significantly indicate the impact of healthcare system dynamics, cultural beliefs, and SRH awareness on men’s decisions to engage with SRH services. Knowledgeable men are more likely to utilize SRH services than less knowledgeable men. Findings also indicate that culture no longer hinders their use of SRH services. Men are deterred by HCWs’ bad attitudes, the unavailability of medicines, and the long distances they must travel to reach health facilities. Furthermore, most barely used SRH services, such as vasectomy, PC screening, and fertility services, are the same services men are less knowledgeable and aware of. We believe that this study has comprehensively investigated SRH issues to comprehend men’s utilization of SRH services. The findings of this study may influence health policymakers and relevant stakeholders to develop policies that strengthen the provision of SRH services for men since their health is significantly compromised by complications resulting from preventable and treatable conditions.

Despite the many educational campaigns on various SRH services in the past, the government still needs to roll out sustained ongoing campaigns at schools, workplaces, communities, and during weekends, as lack of awareness remains a challenge. Information needs to demystify myths and misconceptions linked to using SRH services. The provision of SRH service information should focus mostly on rural areas since SRH information is often disseminated by health pharmacists, public health practitioners, doctors, nurses, and community health workers, who are predominantly situated in urban areas. Information, programs, and services must be male-centered and easily accessible on TV, social media, in magazines, and local newspapers. The government needs to add more male HCWs since most men perceive it as an invasion of privacy to undress or discuss their SRH conditions in front of female nurses during consultations with doctors. Strengthening the relationship between traditional healers and medical doctors can improve patient management. Waiting times still need more improvement because men would normally consider visiting traditional healers and pharmacies as they claim to receive help quicker.

## 5. Limitations

This study only covered certain districts in KZN and focused solely on men’s views without considering perspectives from other SRH stakeholders. Further research should explore stakeholder opinions on SRH service utilization to encourage regular male access. While subjective, investigations into individuals’ sexual behavior, attitudes, and experiences are crucial. The generalization of the findings to other geographic regions may be limited since the study settings were dominated by a few South African ethnic groups, and most of the participants are from poor socio-economic backgrounds. Investigating individuals’ healthcare-seeking behavior, attitudes, and experiences is subjective. Therefore, men are more likely to provide socially acceptable responses. Men are often perceived as strong and independent, which may result in reluctance or embarrassment to discuss health issues and behaviors.

## 6. Conclusions

This study highlighted the perceived facilitators and barriers to utilizing SRH services, which have policy and programmatic implications. Men are still reluctant to utilize SRH services due to poor healthcare provision, particularly in the public sector. The findings indicate the importance of catering to men, especially from disadvantaged areas of society, regardless of culture, religion, and ethnicity. The provision of SRH service information should focus mostly on rural areas because most SRH information is often disseminated by health pharmacists, public health practitioners, doctors, nurses, and community health workers, who are predominantly situated in urban areas. To encourage more men to access SRH services, the health sector must become more male-friendly, particularly in rural areas and facilities with mostly female staff. Barriers like men’s lack of awareness, socio-economic issues, negative staff attitudes, and system inadequacies will persist unless addressed. Despite the many educational campaigns on various SRH services in the past, governments still need to roll out sustained ongoing campaigns at schools, workplaces, communities, and during weekends, as lack of awareness remains a challenge. Furthermore, some men still believe in traditional medicine. Therefore, establishing traditional and medical health practitioners’ forums in the province needs to be strengthened to enforce relations and proper coordination of SRH services. Health policymakers and relevant stakeholders need to pay attention to the provision of SRH services for men because men’s health is significantly compromised by complications resulting from preventable and treatable conditions. Policymakers should also focus on men’s SRH needs by improving services by hiring more male nurses, ensuring adequate medical supplies, reducing wait times, and enhancing health education and awareness campaigns.

## Figures and Tables

**Table 1 ijerph-21-01632-t001:** Focus Group Discussion questionnaire on the utilization of sexual and reproductive health services by men (*n* = 72).

1. What sexual and reproductive health issues do you know? 2. What do you think are the major sexual and reproductive health problems facing males in KZN today? 3. What are some of the most common health issues that make men visit reproductive health services? 4. Are the health-related issues different for males and females? If so, what is the difference? 5. When you have health problems, where else do you go for help besides clinics and hospitals? What are your reasons? ________________________ 6. As far as you know, is there something being implemented by the government, nongovernmental, or other organizations to address men’s sexual health problems? 7. What else could the government and other stakeholders do to meet the sexual and reproductive health needs of males? 8. What else could the government do to support men and to improve the services offered at the clinics? 9. Can you tell me some of the services that you feel should be provided for males? 10. Please tell me about organizations that you know that provide sexual health services around KZN. What kind of services do these organizations provide? __________ 11. What do you think is the reason men do not go to the clinic for sexual and reproductive health services (probe)? What other things do you think discourage males from visiting the clinics or hospitals? 12. What things would you like to be changed at the clinic so that men could be motivated to come? 13. What other services do you think could be helpful in meeting the sexual and reproductive health needs of males? 14. Do you have any problems or difficulties in accessing men’s sexual health services? If ‘yes’, what kind of problems or difficulties do you experience? 15. What do your friends and other community members think of you when you go to a sexual and reproductive health service at your local clinic? 16. If you were to design a sexual and reproductive health service for men, what would you do? What would you prefer as a male? What would you prefer it to look like? What things would you prefer to be in that particular service? What things would you like to be changed at the clinic so that men can be motivated to go there more often when they need the services?	17. Do you have any suggestions about how health facilities could improve services to meet your (men’s) needs? 18. What methods of family planning do you know? 19. Do you approve of the use of family planning within the marriage? Explain your reasons for approval/disapproval of family planning. 20. Have you ever used any modern family planning methods? 21. Have you ever attended a family planning clinic with your partner? 22. Which family planning method do you think is the most effective? 23. Who makes decisions regarding family planning in your marriage/relationship? 24. How many times has family planning been discussed in your marriage/relationship? 25. How can male awareness of family planning be improved? 26. Have you ever attended your local clinic for family planning services? Please share your experience with us. How often do you attend these family planning services? What are the reasons for attending these family planning services? 27. What would discourage you from visiting the clinic for family planning services? 28. What areas would you like to be improved within the present reproductive health services? 29. What would encourage you to visit the clinic for family planning services? 30. Can you recall a positive and negative incident you experienced while visiting the clinic? Has this incident influenced your pattern of attending the clinic? 31. In your own opinion, would you say the services are adequate to meet your current needs? 32. How can you prevent yourself from contracting HIV? 33. Do you believe in counseling? 34. What do you understand about screening? What SRH screenings do you know?

**Table 2 ijerph-21-01632-t002:** Demographic characteristics of the respondents (n = 72).

Explanatory Variable	Categories of Explanatory Variables	Number of Participants	Participants’ Codes (Where 1st Row Below: Alphabet-Gr = Participant in ‘Hospital Gr’; Alphabet-L = Participant in ‘Hospital L’; etc. And 2nd Row Below: Alphabet-G = Participant in ‘Hostel G’; Alphabet-J = Participant ‘Hostel J’; etc.)
Respondents	Men (urology clinics)	33	A(35)-Gr; B(61)-Gr; C(36)-Gr; D(79)-Gr; E(59)-Gr; F(62)-Gr; G(85)-Gr; H(44)-Gr; A(77)-L; B(52)-L; C(83)-L; D(56)-L; E(74)-L; F(55)-L; G(45)-L; H(59)-L; A(58)-N; B(75)-N; C(53)-N; D(45)-N; E(75)-N; F(68)-N; G(78)-N; H(41)-N; A(69)-S; B(55)-S; C(79)-S; D(45)-S; E(52)-S; F(24)-S; G(31)-S; H(32)-S; I(29)-S
	Men (hostels)	39	A(71)-G; B(41)-G; C(32)-G; D(69)-G; E(45)-G; F(37)-G; G(48)-G; H(82)-G; I(38)-G; A(45)-J; B(39)-J; C(44)-J; D(40)-J; E(51)-J; F(32)-J; G(27)-J; H(52)-J; I(47)-J; A(54)-K; B(52)-K; C(68)-K; D(79)-K; E(58)-K; F(72)-K; G(59)-K; H(57)-K; I(48)-K; A(50)-W; B(45)-W; C(39)-W; D(33)-W; E(31)-W; F(42)-W; G(32)-W; H(49)-W; I(29)-W; J(35)-W; K(56)-W; L(51)-W
Age categories	17 to 25 years (youthful)	1	F(24)-S
	26 to 44 years (young)	22	B(41)-G; C(32)-G; F(37)-G; I-G(28); B(39)-J; C(44)-J; D(40)-J F(32)-J; G(27)-J; C(39)-W; D(33)-W; E(31)-W; F(42)-W; G(32)-W; I(29)-W; J(35)-W; A(35)-Gr; C(36)-Gr; H(44)-Gr; G(31)-S; H(32)-S; I(29)-S
	45 to 60 years (middle-aged	30	E(45)-G; G(48)-G; A(45)-J; E(51)-J; I(47)-J; H(52)-J; A(54)-K; B(52)-K; E(58)-K; G(59)-K; H(57)-K; I(48)-K; A(50)-W; B(45)-W; L(51)-W; H(49)-W; K(56)-W; E(59)-Gr; G(45)-L; B(52)-L; D(56)-L; F(55)-L; H(59)-L; A(58)-N; D(45)-N; H(41)-N; C(53)-N; B(55)-S; D(45)-S; E(52)-S
	61 to 75 years (elderly)	11	A(71)-G; D(69)-G; C(68)-K; F(72)-K; B(61)-Gr; F(62)-Gr; E(74)-L; B(75)-N; E(75)-N; F(68)-N; A(69)-S
	75 to 90 years (senile)	8	H(82)-G; D(79)-K; D(79)-Gr; G(85)-Gr; A(77)-L; C(83)-L; G(78)-N; C(79)-S
Place of stay	Urban	10	F(62)-Gr; H(44)-Gr; B(52)-L; G(45)-L; H(59)-L; B(75)-N; A(69)-S; F(24)-S; H(32)-S; I(29)-S
	Township	12	B(61)-Gr; E(59)-Gr; G(85)-Gr; D(56)-L; F(55)-L; A(58)-N; C(53)-N; E(75)-N; F(68)-N; D(45)-S; E(52)-S; G(31)-S
	Rural	11	A(35)-Gr; C(36)-Gr; D(79)-Gr; A(77)-L; C(83)-L; E(74)-L; D(45)-N; G(78)-N; H(41)-N; B(55)-S; C(79)-S
	Hostel	39	A(71)-G; B(41)-G; C(32)-G; D(69)-G; E(45)-G; F(37)-G; G(48)-G; H(82)-G; I(38)-G; A(45)-J; B(39)-J; C(44)-J; D(40)-J; E(51)-J; F(32)-J; G(27)-J; H(52)-J; I(47)-J; A(54)-K; B(52)-K; C(68)-K; D(79)-K; E(58)-K; F(72)-K; G(59)-K; H(57)-K; I(48)-K; A(50)-W; B(45)-W; C(39)-W; D(33)-W; E(31)-W; F(42)-W; G(32)-W; H(49)-W; I(29)-W; J(35)-W; K(56)-W; L(51)-W
Marital status	Single	39	A(71)-G; C(32)-G; E(45)-G; F(37)-G; G(48)-G; I(38)-G; A(45)-J; B(39)-J; C(44)-J; D(40)-J; F(32)-J; G(27)-J; H(52)-J; I(47)-J; A(54)-K; H(57)-K; I(48)-K; B(45)-W; C(39)-W; D(33)-W; E(31)-W; F(42)-W; G(32)-W; H(49)-W; I(29)-W; J(35)-W; K(56)-W; A(35)-Gr; C(36)-Gr; B(52)-L; F(55)-L; H(59)-L; D(45)-N; F(68)-N; H(41)-N; F(24)-S; G(31)-S; H(32)-S; I(29)-S
	Married	19	B(41)-G; D(69)-G; E(51)-J; B(52)-K; D(79)-K; G(59)-K; A(50)-W; L(51)-W; F(62)-Gr; H(44)-Gr; A(77)-L; C(83)-L; G(45)-L; A(58)-N; B(75)-N; C(53)-N; A(69)-S; B(55)-S; D(45)-S
	Divorced	7	C(68)-K; E(58)-K; B(61)-Gr; E(59)-Gr; D(56)-L; E(75)-N; E(52)-S
	Widowed	7	H(82)-G; F(72)-K; D(79)-Gr; G(85)-Gr; E(74)-L; G(78)-N; C(79)-S
Educational level	Pre-secondary school education	46	I(38)-G; H(82)-G; E(45)-G; A(71)-G; D(69)-G; A(45)-J; E(51)-J; I(47)-J; A(54)-K; B(52)-K; C(68)-K; D(79)-K; E(58)-K; F(72)-K; G(59)-K; H(57)-K; A(50)-W; C(39)-W; E(31)-W; F(42)-W; L(51)-W; K(56)-W; B(61)-Gr; C(36)-Gr; D(79)-Gr; E(59)-Gr; F(62)-Gr; G(85)-Gr; H(44)-Gr; A(77)-L; C(83)-L; D(56)-L; E(74)-L; F(55)-L; H(59)-L; B(75)-N; C(53)-N; E(75)-N; F(68)-N; G(78)-N; A(69)-S; B(55)-S; C(79)-S; D(45)-S; E(52)-S; H(32)-S
	Secondary school education	20	C(32)-G; F(37)-G; G(48)-G; C(44)-J; D(40)-J; G(27)-J; H(52)-J; I(48)-K; B(45)-W; G(32)-W; I(29)-W; J(35)-W; A(35)-Gr; B(52)-L; G(45)-L; A(58)-N; D(45)-N; H(41)-N; F(24)-S; G(31)-S
	Tertiary education	6	B(41)-G; B(39)-J; F(32)-J; D(33)-W; H(49)-W; I(29)-S
Ethnicity	Zulu	47	A(71)-G; B(41)-G; D(69)-G; F(37)-G; H(82)-G; B(39)-J; C(44)-J; E(51)-J; F(32)-J; H(52)-J; I(47)-J; A(54)-K; B(52)-K; C(68)-K; D(79)-K; E(58)-K; F(72)-K; G(59)-K; H(57)-K; I(48)-K; A(50)-W; B(45)-W; D(33)-W; E(31)-W; F(42)-W; H(49)-W; B(61)-Gr; C(36)-Gr; D(79)-Gr; A(77)-L; C(83)-L; D(56)-L; E(74)-L; F(55)-L; A(58)-N; C(53)-N; D(45)-N; E(75)-N; F(68)-N; G(78)-N; H(41)-N; C(79)-S; D(45)-S; F(24)-S; G(31)-S; H(32)-S; I(29)-S
	Xhosa	11	C(32)-G; E(45)-G; G(48)-G; A(45)-J; D(40)-J; G(27)-J; I(29)-W; J(35)-W; K(56)-W; L(51)-W; A(35)-Gr
	Sotho	5	I(38)-G; C(39)-W; G(32)-W; E(59)-Gr; B(55)-S
	White	2	F(62)-Gr; B(75)-N
	Indian	6	G(85)-Gr; H(44)-Gr; G(45)-L; H(59)-L; A(69)-S; E(52)-S
	Colored	1	B(52)-L
Employment	Employed	34	A(71)-G; B(41)-G; E(45)-G; G(48)-G; I(38)-G; C(44)-J; D(40)-J; E(51)-J; H(52)-J; A(54)-K; B(52)-K; E(58)-K; G(59)-K; A(50)-W; B(45)-W; F(42)-W; G(32)-W; H(49)-W; J(35)-W; A(35)-Gr; B(61)-Gr; E(59)-Gr; H(44)-Gr; D(56)-L; G(45)-L; A(58)-N; D(45)-N; F(68)-N; H(41)-N; B(55)-S; D(45)-S; E(52)-S; F(24)-S; H(32)-S
	Unemployed	38	C(32)-G; D(69)-G; F(37)-G; H(82)-G; A(45)-J; B(39)-J; F(32)-J; G(27)-J; I(47)-J; C(68)-K; D(79)-K; F(72)-K; H(57)-K; I(48)-K; C(39)-W; D(33)-W; E(31)-W; I(29)-W; K(56)-W; L(51)-W C(36)-Gr; D(79)-Gr; F(62)-Gr; G(85)-Gr; A(77)-L; B(52)-L; C(83)-L; E(74)-L; F(55)-L; H(59)-L; B(75)-N; C(53)-N; E(75)-N; G(78)-N; A(69)-S; C(79)-S; G(31)-S; I(29)-S

**Table 3 ijerph-21-01632-t003:** Themes emanating from the participants’ comments.

	THEMES
Barriers to SRH Service Utilization	Lack of awareness of SRH services; Participants’ reluctance to access SRH services; Influence of culture and religion; Lack of financial resources; Influence of relationship dynamics; Perceived low risk of individual sexual behaviors; Healthcare factors that discourage men from accessing SRH services.
Facilitators that encourage SRH service utilization	Healthcare enabling factors; Access to general information on SRH services; Personal motivational factors

## Data Availability

The raw data supporting the conclusions of this article will be made available by the authors on request.

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
