# Peer review of "Exploratory Qualitative Study to Investigate Factors Influencing Men’s Utilization of Sexual and Reproductive Health Services in Kwa-Zulu Natal"

_ijerph, 2024, doi:10.3390/ijerph21121632_

Round 1

Reviewer 1 Report

Comments and Suggestions for Authors

Dear Author:

We have reviewed your article and would like to offer some suggestions for improvement:

1. We feel that the title is too long. We suggest you make it more synthetic to better capture the reader's attention.

2. It is important that acronyms are explained the first time they are used in the text to ensure that they are understood by all readers.

3. Given the descriptive nature of the title, we recommend expanding the theoretical framework and providing a more detailed explanation of the situation being addressed. What do you mean by men living in shelters? Why do they live there? Please help us understand the context of the study.

4. We need more details about the interviews conducted. It would be useful to know how they were structured, how they were designed and how they were validated. We recommend including a table with the questions to facilitate the review of their content, as well as a robust explanation of their construction and validation process.

5. We require more information on how the responses were categorised and how they were analysed to provide validity and objective interpretation of the interviews. This process is not clear in the current article.

6. While the study is interesting, it needs to be more explanatory both in the context of men living in shelters and in the methodology employed. It is crucial to detail the instrument used and how it was interpreted objectively.

7. We need the theoretical and practical implications of the study and its conclusions to be clearly stated. This will help readers understand the relevance and impact of your research.

8. We suggest you update the references, paying special attention to sources from the last 3-4 years to ensure the currency and relevance of the bibliography.

We thank you for your dedication and hope that these suggestions will help you to strengthen your article.

Author Response

Dear Reviewer 

Kindly find the attached Comments and Responses.

Kind regards

Mr. Mpumelelo Nyalela   

Reviewer 2 Report

Comments and Suggestions for Authors

Dear authors, I appreciate the opportunity to read your qualitative design manuscript. I will use the COREQ guideline to evaluate its content.

Title: in my opinion, the title is a bit long (a maximum of 15 words is recommended). It is advisable to include a term that refers to the qualitative design of the research.

Abstract: Avoid using acronyms in the abstract. If an acronym should be used, it should be explained in its first use (for example: SRH). Regarding methodology, more information should be provided on participants, inclusion and exclusion criteria, analysis process and ethical criteria. For the results, it is necessary to be more precise with the results of the participants (number of individuals interviewed) and categories of analysis or themes identified.

Keywords should include MeSH/DeCS terms as far as possible. Avoid unspecific terms (e.g. Men) and include a term for qualitative research. Do not use acronyms for keywords (SHR).

Introduction: explain the acronym UNFPA in its first use. The objective should be described with the same concepts in the abstract and in the text of the manuscript.

Material and methods: should start with the subheading: design, which describes the methodological approach (phenomenology, ethnography, grounded theory...). it makes more sense for settings to be the second subsection. Information on the data collection technique (focus group discussion) should not be intermingled in the design section. This information should be transferred to the data collection subheading. No reference should be made in the methodology to the results (Table 1). The criteria for recruiting informants and whether there were informants who were invited to participate and did not participate in the study should be described. Avoid listing inclusion and exclusion criteria as a checklist. It is better to write them without listing them. Exclusion criteria should not be limited to describing the opposite of inclusion criteria (e.g., men over 15 years of age are included and men under 15 years of age are excluded). Describe if software has been used for data analysis (if not used, this should also be reported). In the section on rigor, a reference author should be cited in this regard (for example: Guba and Lincoln).

Results: It is not correct to start the results with table 1 (first the information in the text and then the table). It is necessary to introduce the table with text in the manuscript. In qualitative studies it is more relevant to prepare tables describing the characteristics of each participant instead of frequencies or percentages. This information is relevant for later results with the extracted verbatims (e.g., Mr. C (44), does he belong to a urology clinic or hostel, what is his age, place of residence, marital status, educational level, ethnicity, employment? Table 2 should be cited in the text and shown after describing the information in the text.

Limitations should be included at the end of the discussion (in the same section).

The conclusions should be included in a specific section. Recommendations should be called implications for clinical practice (they may be included in the conclusions, but not at the head of the section).

References: review the style and add the link to the primary source or DOI when appropriate.

Author Response

Dear Reviewer 

Kindly find the attached Comments and Responses

Kind regards 

Mr. Mpumelelo Nyalela 

Round 2

Reviewer 1 Report

Comments and Suggestions for Authors

Thank you for the opportunity to review the revision of your manuscript. Below are some specific points that could strengthen your work in addition to what has already been done:

1. It is important to explain how the scripts used in the focus groups were validated. For example, it would be useful to detail whether subject matter experts were involved in the question selection process or whether the questions were developed solely by the authors. Clarity in the development of the interviews is crucial. If the authors had the participation of specialists, please include this information and explain how their input was incorporated into the instrument.

2. I suggest you expand the analysis of how KwaZulu-Natal-specific cultural values influence men's decisions regarding sexual and reproductive health. This could include specific examples or a more in-depth analysis of the cultural and religious beliefs mentioned in the text.

3. Link the findings to previous studies to highlight similarities and differences, highlighting how this study contributes to existing knowledge.

4. Emphasize how your suggestions can be implemented in the local context and assess their applicability to other similar regions.

I am confident that these suggestions will help strengthen your manuscript and increase its scholarly and practical impact.

Author Response

Dear Reviewer 

Please find attached the responses to the comments and the manuscript with track changes. 

Kind regards 

Mr Mpumelelo Nyalela 
